# Autoptic Findings in Cases of Sudden Death Due to Kawasaki Disease

**DOI:** 10.3390/diagnostics13111831

**Published:** 2023-05-23

**Authors:** Giacomo Visi, Federica Spina, Fabio Del Duca, Alice Chiara Manetti, Aniello Maiese, Raffaele La Russa, Paola Frati, Vittorio Fineschi

**Affiliations:** 1Department of Surgical Pathology, Medical, Molecular and Critical Area, Institute of Legal Medicine, University of Pisa, 56126 Pisa, Italy; visi.giacomo92@gmail.com (G.V.); f.spina6@studenti.unipi.it (F.S.); 2Department of Anatomical, Histological, Forensic and Orthopedic Sciences, Sapienza University of Rome, Viale Regina Elena 336, 00161 Rome, Italy; fabio.delduca@uniroma1.it (F.D.D.); paola.frati@uniroma1.it (P.F.); vittorio.fineschi@uniroma1.it (V.F.); 3Department of Public Health and Infectious Diseases, Sapienza University, 00185 Rome, Italy; alicechiara812@gmail.com; 4Department of Clinical and Experimental Medicine, Section of Forensic Pathology, University of Foggia, 71122 Foggia, Italy

**Keywords:** Kawasaki disease, postmortem investigation, histology, sudden death

## Abstract

Kawasaki disease (KD) is the second-most-common childhood vasculitis, and its etiology is still unknown today. Even though the acute illness is usually self-limiting, sometimes, it can generate complications, such as coronary artery aneurysms (CAA), acute myocardial infarction (AMI), heart failure, or arrhythmias, and can rarely cause sudden or unexpected deaths. We present a review of the literature, which collects autoptic and histopathological data relating to many of the cases of these deaths. On the basis of the titles and abstracts, we selected 54 scientific publications for a total of 117 cases. Among them, as expected, the majority of the deaths were due to AMI (41.03%), arrhythmia (8.55%), acute coronary syndrome (8.55%), and CAA rupture (11.97%), involving mostly 20-year-olds or younger individuls (69.23%). This is not surprising since the CAs are the most involved arteries. Gross autoptic and histopathological findings are reported in the paper. Our work revealed that, when compared with the incidence of KD, only a few cases suffered from sudden death, underwent an autoptic examination, and were then described in the literature. We suggest that researchers should perform autopsies to gain a better understanding of the molecular pathways involved in KD so as to propose further innovative therapeutic protocols or implement more appropriate prevention schemes.

## 1. Introduction

Kawasaki disease (KD), or mucocutaneous lymph node syndrome, is an acute, self-limited vasculitis of medium-sized vessels, affecting mainly the coronary arteries. It mostly occurs during childhood [1,2]. and it is the second-most-common childhood vasculitis. As a matter of fact, it is second only to immunoglobulin A (IgA) vasculitis (Henoch–Schönlein purpura) [3]. It was discovered by Dr. Tomisaku Kawasaki, who studied the first case in 1961: A 4-year-old boy unusually presenting with rash and fever. In 1967, he published his first study, conducted on 50 similar cases, which brought him to a definition of this pathology. His study was first published in Japanese [4] and subsequently translated into English in 1974 [5]. 

Although many years have passed since Dr. Kawasaki’s first study, the KD etiology is still unknown today, but several theories have been proposed [6]. The hypothesis that it is caused by a transmissible agent (e.g., Parvovirus B19 [7], Propionibacterium [8], Human Bocavirus [9]) is the most debated. As a matter of fact, it has been proposed that bacterial toxins or viral agents, acting as superantigens, can let an inflammatory cascade begin. The inflammatory cascade, which primarily affects the walls of the coronary vessels, leads to detectable macroscopic and microscopic alterations, such as coronary artery (CA) thrombosis, CA stenosis, coronary artery aneurysm (CAA), CA wall inflammatory infiltrates, and myocardial infarction. These changes are responsible for the onset of the KD signs and symptoms [10]. Other possible etiological causes include genetic [11,12], immune [13,14,15,16,17], and environmental factors [18]. 

According to the Diagnostic Guidelines for Kawasaki Disease [19], the principal symptoms are persisting fever (5 days or more), bilateral conjunctival congestion, oral changes, polymorphous exanthema, peripheral extremities changes, and cervical lymphadenopathy. The KD diagnosis is established by the presence of at least 5 days of persisting fever and 4 of the other principal criteria when any other explanation for the illness is not possible. The classic diagnostic criteria are not always fulfilled by all patients with KD. Some of them present with a fever and less than 4 of the other criteria and are then referred to as “incomplete” or “atypical” KD patients. Diagnosis may be difficult in such cases.

Acute KD illness is usually self-limiting, but unfortunately, in some cases, it can lead to short-term as well as medium- and long-term complications [20,21,22]. Among these, there are coronary artery aneurysms (CAA), acute myocardial infarction (AMI), heart failure, arrhythmias, and hemodynamic instability, characterizing a condition known as Kawasaki disease shock syndrome (KDSS) [23].

The advent of intravenous immunoglobulin (IVIG) therapy brought about a consistent decrease in the KD mortality rate [24]. 

Nowadays, the rare episodes of death related to KD are often due to serious involvement of the cardiovascular system with consequent AMI, arrhythmias, or coronary artery aneurysm rupture (CAA rupture). Among the factors that can increase the risk of the occurrence of acute fatal events, the most representative are the size of the coronary aneurysm and stenosis of the lumen. Therefore, we present a review of the literature, which collects data related to many cases of sudden and unexpected death from KD. Particular attention was paid to autopsy and histological data.

## 2. Materials and Methods

The present systematic review was carried out according to the Preferred Reporting Items for Systematic Review (PRISMA) standards [25]. A systematic literature search and critical review of the collected studies were conducted. An electronic search of PubMed, Science Direct Scopus, Google Scholar, and Excerpta Medica Database (EMBASE) was performed from database inception until February 2023. The search terms were “Kawasaki disease + sudden death”, “Kawasaki disease + fatal”, “Kawasaki disease + sudden cardiac death”, “Kawasaki disease + sudden infant death”, “Kawasaki disease + unexpected death”, “Kawasaki disease + autopsy”, “mucocutaneous lymph node syndrome + sudden death”, “mucocutaneous lymph node syndrome + fatal”, “mucocutaneous lymph node syndrome + sudden cardiac death”, “mucocutaneous lymph node syndrome + sudden infant death”, “mucocutaneous lymph node syndrome + unexpected death”, and “mucocutaneous lymph node syndrome + autopsy”, and these terms were searched in the titles, abstracts, and keywords. The bibliographies of all located papers were examined and cross-referenced to further identify relevant literature. A methodological appraisal of each study was conducted according to the PRISMA standards, including an evaluation of bias. The data collection process included study selection and data extraction. The following inclusion criteria were used: (1) original research articles, (2) reviews and mini-reviews, (3) case reports or case series, (4) only papers written in English, (5) only cases of sudden or unexpected death related to KD, (6) cases where the cause of death was certainly attributable to KD, and (7) only cases affected by primary KD. Non-English papers, papers regarding cases of survived patients, papers in which the cause of death was not directly attributable to KD, papers in which the subject had secondary KD, and papers presenting cases where death occurred after a long hospital stay (>3 days) following the acute event were excluded. The latter were excluded, because the purpose of the review is to highlight the main anatomical and histopathological findings on sudden or unexpected deaths due to Kawasaki. A long hospital stay was therefore considered a possible source of bias in data interpretation. Two researchers (G.V. and F.S.) independently examined the papers with titles or abstracts that appeared to be relevant and selected those that analyzed sudden or unexpected deaths due to KD. Disagreements concerning eligibility among the researchers were resolved by consensus. Preprint articles were included. Data extraction was performed by two investigators (A.C.M. and A.M.) and verified by other investigators (G.V., F.S. and F.D.D.). This study was exempt from institutional review board approval, as it did not involve human subjects.

## 3. Results

The search identified 216 articles, which were screened to exclude duplicates. The resulting 152 references were then screened based on the titles and abstracts of the articles. This procedure left 88 articles for further consideration. These publications were carefully evaluated, considering the main aims of the review. This evaluation left 54 scientific papers comprising original research articles, case reports, and case series. 

Figure 1 illustrates our search strategy.

We found a total of 117 cases of sudden or unexpected death related to KD in the literature. In Table 1, brief descriptions of the studies included in this review are reported.

Autoptic and histopathological findings were reported for 67 and 72 cases, respectively. As shown in Table 2, which summarizes the main characteristics of the cases included in this review, there were 86 males (73.50%), 20 females (17.09%), and for 11 people, gender was not specified (9.40%). The age of the subjects was specified in only 102 out of the 117 cases. Among them, the most affected age group was the <20-year-old population (69.23%). Out of these, 28 subjects (23.93%) were <5 years old, 26 (22.22%) were 6–12 years old, and 27 (23.08%) were 13–20 years old. Age was not available in 15 cases (12.82%). 

A total of 80 subjects had an in-life diagnosis. Among them, 37 were treated (31.63%), 7 were not treated (5.98%), and no treatment data were available for 36 of them (30.77%). Thirty-seven cases had a postmortem diagnosis (31.62%).

As shown in Table 3, there were 48 deaths due to AMI (41.03%), 7 deaths due to acute cardiac failure (ACF) (5.98%), 2 deaths due to pericarditis (1.71%), 2 deaths due to pancarditis (1.71%), 6 deaths due to myocarditis (5.13%), 10 deaths due to arrhythmia (8.55%), 10 deaths due to acute coronary syndrome (8.55 %), 14 deaths due to CAA rupture (11.97 %), 1 death due to cardiac tamponade (0.85%), 1 death due to acute encephalitis (0.85%), 1 death due to cerebral hemorrhage (0.85%), 1 death due to cerebral hypoxia (0.85%), and 14 deaths with undefined causes (11.97%). A pie chart on the causes of death is presented in Figure 2.

As previously stated, autopsy investigations and histological examinations were reported for 67 and 72 cases, respectively. In Table 4, a summary of the main gross findings is shown. Among the most frequent findings, out of 65 cardiac death cases, 19 presented with an enlarged heart, 21 had signs of myocardial fibrosis, and 18 had coronary stenosis. Moreover, CAAs affected 46 people, and CAA rupture was reported in 14 cases.

Among the most frequent microscopic findings, out of 72 cardiac death cases, AMI was reported in 17 cases, myocardial fibrosis was reported in 17 cases, coronary artery stenosis was reported in 16 cases, coronary artery thrombosis was reported in 19 cases, coronary artery wall inflammatory infiltrates were reported in 26 cases, and CAA thrombosis was reported in 17 cases. In Table 5, a summary of the main microscopic findings is reported.

## 4. Discussion

Our review identified 117 cases of sudden death from KD in the literature. Unfortunately, only some of them have been completely described (67 and 72 with autoptic and histopathological reports, respectively). The first article about KD dates back to 1967, with the report of an acute febrile mucocutaneous syndrome with lymph node enlargement and cutaneous manifestations [4,80]. In almost 60 years, a lot has been discovered about this peculiar disease, and significant progress has been made in terms of diagnosis, prevention, treatment, and follow up of the disease. Indeed, only a few autoptic reports are present in recent literature, also due to the fact that the pathology is, to date, rarely fatal. Nowadays, it is well-known that autopsy is a fundamental tool to understand the physiopathology of diseases better and deeply [81]. As a matter of fact, autoptic data have been fundamental to provide essential information about the molecular pathways of pathological features, allowing targeted therapies and specific diagnostic protocols [82].

The present systematic review revealed that, when compared with the incidence of KD [83,84], only a few cases suffered from sudden death, underwent an autoptic examination, and were then described in the literature, mainly through case reports or series. 

Ayusawa et al. described cases of sudden death among students between 1990 and 1999 and then between 2000 and 2009, in Japan [79]. Their research is very interesting, because it is one of the largest case series. The authors did not provide any autoptic data, meaning that these findings are still not considered to be fundamental for understanding a disease. However, their studies seem to show a reduction in sudden death cases of about 50% for all new diagnoses of KD and a reduction of approximately 80% in patients also affected by CAA. This may be due to the increased knowledge about KD, which allows earlier diagnosis and better in-life treatment. Suda et al. stated that the introduction of anticoagulant therapy, particularly warfarin, helps to reduce the incidence of myocardial infarction [85]. 

Two decades after its discovery, KD surpassed rheumatic fever as the most common cause of acquired heart disease in children in Asia [86]. As a matter of fact, we identified 81 cases of death due to KD among people aged younger than 20 years, which represents 69.23% of all identified cases, and most of these cases involved Asian subjects.

Furthermore, an aspect of non-negligible importance is the psychological impact that the disease has on these young subjects. As reported by Shain B.N., adolescents living with chronic disease have a greater risk of suicide attempts [87]. King et al. showed that kids affected by KD are more likely to present with social problems and experience anxious–depressed behavior and attention difficulties compared with their healthy siblings [88].

Further studies [89,90] support the findings about the negative impact of KD on behavior. Alves et al. showed that 20% of these patients experience frequent irritability, aggressiveness, attention deficit, learning deficit, and antisocial behaviour [90]. A central nervous system pathology related to cerebral vasculopathy has been taken into consideration to explain a possible physiopathology of the problem [89].

Kato et al. studied the natural course of coronary lesions in KD patients, suggesting that the cause of intimal proliferation and obstruction could be vascular remodeling [91]. 

Burns et al. and Sasaguri and Kato, in their morphologic postmortem studies related to the acute phase of the disease, showed that intimal obstruction is caused by chronic and persisting activity of inflammatory infiltrates on the endothelium and artery wall. Such inflammatory activity, associated with remodeling activity, results in aneurysm, thrombosis, or intimal proliferation, similar to atherosclerotic coronary disease [92,93]. Nakamura et al. followed up 6576 patients with KD after their first medical examination. At the end of the observational study, both deceased patients and still-alive patients were taken into consideration for the data analysis. The study found out that the mortality rate did not increase significantly after the acute phase of the disease [94].

In this paragraph, we present a detailed overview of the main molecular mechanisms involved in the pathology. In its acute phase, KD is characterized by abnormal activation of immunocompetent cells such as monocytes/macrophages and lymphocytes, leading to the secretion of inflammatory cytokines and chemokines, such as tumor necrosis factor α (TNFα), interferon-γ (IFN-γ), interleukins (e.g., IL-6), and monocyte chemoattractant protein (MCP)-1. These molecules activate immune cell interactions and the endothelial cells, causing, on the surface of the latter, the expression of adhesion molecules. As a result, the leucocytes adhere firmly to them and, eventually, migrate through the vascular walls. A vasculitis process hence begins, and endothelial cells and smooth muscle cells are damaged. Additionally, various vasoactive substances, including growth factors, such as vascular endothelial growth factor (VEGF) and platelet-derived growth factor (PDGF), as well as endothelin and nitric oxide, are produced, allowing the induction of migration and the proliferation of intimal smooth muscle cells. Moreover, these substances lead to an increase in vascular permeability, causing coronary artery dilation [6].

The histological findings collected through this review corroborate this interpretation, showing that the arterial wall is involved in chronic and persistent inflammatory activity. As a matter of fact, while limited by a small number of cases, our data show CA wall inflammatory infiltrates and various degree of perivascular myocardial inflammation in most cases, as well as systemic vessel inflammatory infiltrates. Inflammation is mainly supported by lymphocytes and granulation tissue.

Tsuda et al. showed that the left coronary artery is the most affected and that the aneurysm size is one of the major predictive factors for the development of myocardial infarction [95].

As expected, the most common cause of death, in the cases reported in this review, involved the heart, presenting as sudden or unexpected cardiac death. This is not surprising, since the CAs are the most involved arteries [96,97]. 

Early sudden cardiac death may end a clinically silent course of disease or deteriorate a symptomatic clinical spectrum, such as AMI, arrhythmia, or heart failure [98]. Although sudden death mainly affects children and adolescents, it can also occur 10 to 20 years after diagnosis, making KD unsafe at any age [99].

The chronic evolution of CAA caused by KD may develop in two ways: it may either resolve through intimal thickening or progress to lumen enlargement without remodelling, leading to acute thrombosis at a later stage [35,53]. CAA rupture is considered to be an exceptional complication of KD [58].

Some authors found that treatment with IVIG combined with acetylsalicylic acid (ASA) is an effective strategy to prevent CA lesions [100,101]. 

Newburger et al. recalled that some clinical trials performed in the 1980s showed a possible reduction in CAA formation from 25% to 5%, thanks to the use of high doses of IVIG and ASA, if administered within the first 10 days after fever onset [102]. The efficacy of high-dose IVIG treatment for acute-stage KD has become widely recognized [103], and today it is administered to 85% of children with acute-stage KD [6]. Simultaneously, IVIG therapy has led to a decrease in the mortality rate from KD by reducing the formation of lesions that affect the CA [6].

However, it should be reported that, although IVIG infusion is an effective treatment for KD, some patients (4%) still develop CAAs [104].

Our review shows that only 31.62% of sudden deaths (37 cases) did not have an ante-mortem diagnosis. This means that in the remaining 68.38% (80 cases), the diagnosis was performed during life. Among them, seven patients (5.98%) were not under treatment for KD. Unfortunately, we do not know if patients received treatment in 36 cases (30.77%). This lack of information in the papers included in this review does not allow us to further discuss the correlation between the incidence of sudden death and in-life treatment or the possibility that a proper therapy program could significantly prevent sudden death in such patients. 

Another aspect that needs to be highlighted is that most of the cases of sudden death reported in the literature were males. Indeed, 73.50% of the cases included in this review were males. This is in accordance with previous research, which showed that KD is more common in boys, with a male:female ratio of about 1.6:1 [105,106]. Furthermore, according to Nakamura et al., males suffering from cardiac complications caused by KD experience a higher mortality rate in comparison to the general population, whereas the mortality rates for females with complications and for both males and females without complications were not increased [94].

The present study is limited by the small number of included cases, which did not allow for a relevant statistical analysis. In particular, it was not possible to compare differences by country or ethnicity. Moreover, there is no uniformity among papers concerning the demographic information provided, and this is another limit of the current study. The scarcity of sudden or unexpected death cases in KD patients, despite its significance as a public health issue, particularly among young people, may be attributed to the perception that an autopsy is unnecessary when the diagnosis has already been established. However, we suggest that researchers should also perform autopsies when an in-life diagnosis is present in order to share their important findings. Postmortem investigations could provide precise information about the molecular mechanisms causing the alterations of the vascular walls and the specific pathophysiology that causes death in those patients. Additionally, conducting autoptic investigations could significantly improve the understanding of both the molecular mechanisms behind the disease as well as the effectiveness of the various treatment options. Thankfully, as mentioned earlier, the number of deaths related to KD has decreased in the past several decades. As a result, it would be beneficial to thoroughly investigate any rare case of fatality in order to apply new histopathological investigation techniques and other diagnostic examinations such as the post-mortem computed tomography [107]. 

## 5. Conclusions

KD mostly occurs during childhood and is the second-most-common childhood vasculitis. Even though the acute illness is usually self-limiting, it can sometimes generate severe complications, such as CAA, AMI, heart failure, arrhythmias, and hemodynamic instability. These complications can be life threatening and can sometimes lead to sudden death. The advent of IVIG therapy brought a consistent decrease in KD complications and, consequently, the mortality rate. 

Despite a high incidence of KD in society, particularly in Asian societies, the scientific papers reviewed in this study indicate that there is not a significant number of documented cases of sudden death resulting from KD in the literature. Moreover, the present systematic review reveals that only a few cases underwent an autoptic examination. However, from our findings it is evident that cardiac involvement was present in almost all cases and that most sudden or unexpected deaths were attributable to AMI, acute coronary syndrome, arrhythmia, or CAA rupture. Little histopathological data are reported in the literature on this topic, but our results underline that the arterial wall is involved in chronic and persisting inflammatory activity in almost all the cases.

Although the number of cases examined was limited, this study aimed to encourage further research on postmortem investigations. As a matter of fact, only through this type of research can more detailed and precise information be obtained to better understand and investigate the molecular pathways that cause alterations in the main vascular walls. The ultimate goals of this new knowledge could include the development of innovative therapeutic protocols and the implementation of more appropriate prevention schemes.

## Figures and Tables

**Figure 1 diagnostics-13-01831-f001:**
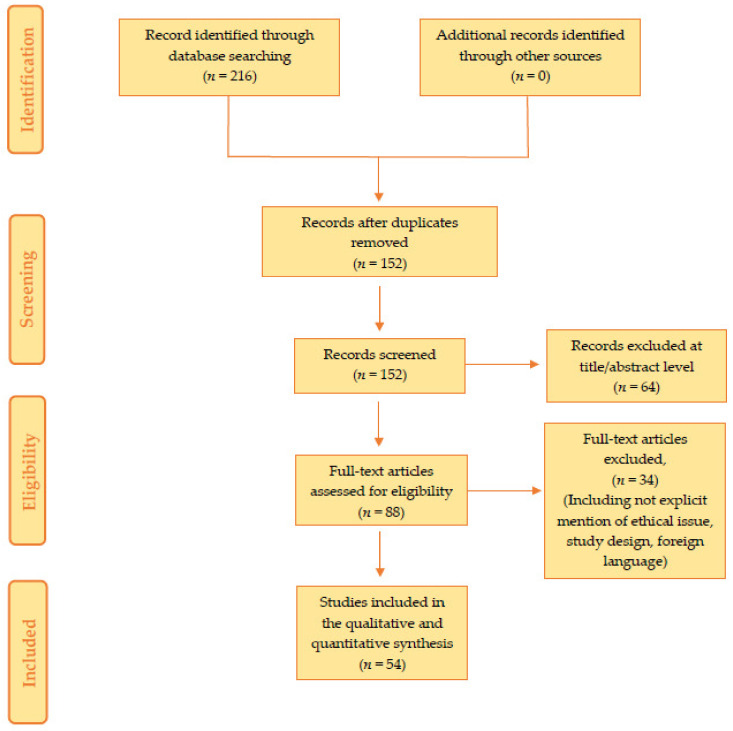
Methodology of the search strategy: We identified 152 articles; screening based on their abstracts left 88 studies, and after a careful evaluation based on the aims of this review, 54 research articles were included.

**Figure 2 diagnostics-13-01831-f002:**
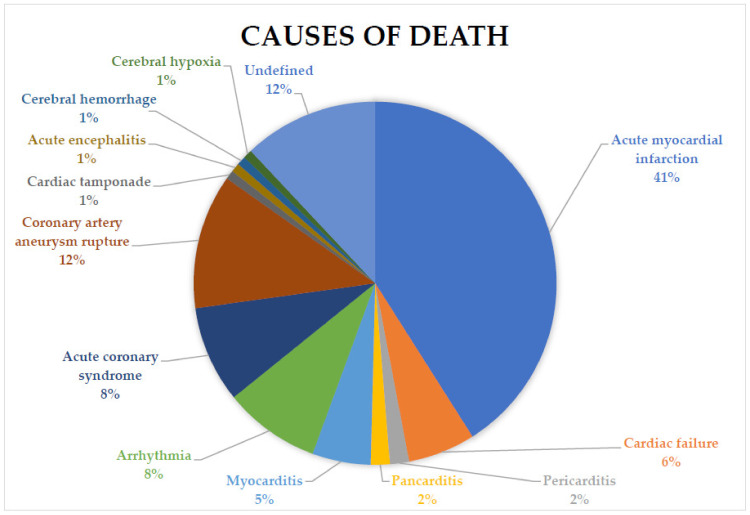
The pie chart shows the percentages of the different causes of death in the 117 analyzed cases.

**Table 1 diagnostics-13-01831-t001:** This table shows brief descriptions of all included cases as well as gross and histological fundings (if reported).

Ref.	Num.	Age and Sex	In-Life/Postmortem Diagnosis and Treatment	Brief Case Description	Autopsy Data	Histological Analysis	Cause of Death
Aterman et al. 1977 [26]	5	4 m.o. M	Postmortem diagnosis.	Hospitalized with BT of 38 °C, respiratory distress, and signs of congestive heart failure, he suddenly died from cardiovascular collapse.	Heart: considerably enlarged. Several CAAs, rounded or fusiform, of variable size and filled with thrombi. Other viscera: marked congestion.	Heart: the CAA wall showed fibrous tissue with loss of smooth muscle and elastic fibers. Fresh and organized thrombi within the CAA lumen. Extensive loss of muscle mass with eosinophilic necrosis, disappearance of fibrils, and a scanty mononuclear response in the myocardium.	ACF
		5 m.o. M	Postmortem diagnosis.	Hospitalized for fever up to 40 °C, not responding to penicillin. During hospital stay, he suffered acute pericarditis and died suddenly.	Heart: fibrinous exudate in the pericardial sac. CAAs of all proximal segments.	CAs showed thinning and destruction of the media, inflammatory infiltration, marked periarteritis, focal fibrinoid changes, and the presence of fibroblasts. Thrombi in the lumen of the arteries. Other vessels: arteritis in the renal arteries and in some branches of the hepatic, peritesticular, and mesenteric arteries.	Pericarditis
		5 m.o. M	Postmortem diagnosis.	Two months after an upper respiratory infection, he underwent intermittent congestive heart failure.	Heart: enlarged with hypertrophied LV. RCA aneurysm.	Heart: CAAs showed severe intimal fibrous thickening and thrombosis with recanalization. Small foci of fibrosis in the myocardium. Kidneys: obliterative changes in the renal arteries and recent infarct in the right kidney.	ACF
		2 y.o. F	Postmortem diagnosis.	Hospitalized for fever and cervical lymphadenopathy. During the hospital stay, the patient developed heart failure and suddenly died.	Heart: fibrinous exudate in the pericardial sac. Dilated and thrombosed CAAs (located in the LCA, LAD, and RCA).	Heart: CAAs showed focal thinning and necrosis of the walls and thrombi in the lumen. CAs had focal acute vasculitis, chronic inflammation with destruction of the media, and fresh proliferating tissue. Focal eosinophilia and a few areas of frank necrosis with neutrophils in the myocardium.	ACF
		3 y.o. M	Postmortem diagnosis.	The patient suddenly died from cardiac arrest after being hospitalized for grunting respirations, bronchial breathing, rales, and rhonchi.	Heart: dilatation of both ventricles and the right atrium. CAAs (located at the LAD and one CX branch).	Heart: CAAs showed fibrous tissue in the wall and fresh thrombi in the lumen. Severe uniform thickening and subsequent marked narrowing of the lumen of the CAs. Markedly scarred myocardium and myocardial infarction with coagulation necrosis and demarcation by neutrophils. Other arteries: severe intimal fibrous scarring, splitting of the internal elastic tunica, and focal scarring of the media with occasional calcification. The abdominal aorta wall had severe fibrous scarring of the intima and media with focal calcium deposition.	AMI
Kegel et al. 1977 [27]	1	12 y.o. M	Postmortem diagnosis.	The patient was found dead in bed 1 week before a scheduled bypass surgery. One day before death, 2 h after playing soccer, he experienced vomiting and chest pain.	Heart: all chambers were dilated. Marked dilatation of the LV. The LV apex was tinged with fibrosis. Numerous pericardial adhesions. Severe sclerosis of the endocardium. The CAA of the LCA contained a thrombus. The LAD, CX, and RCA were thickened and the lumens were narrowed with complete obstruction of the proximal RCA.	Heart: the CAAs demonstrated thickening of the media, subintimal fibrosis with marked hyalinization, and extensive calcification, and the lumen contained a recent thrombus. Severe nuclear hypertrophy and fibrosis in the myocardium. The LV septum demonstrated severe endocardial sclerosis.	AMI
Wilson et al. 1979 [28]	1	4 y.o. M	In-life diagnosis. Treated with ASA.	Sudden death of a child who was hospitalized after three weeks of cervical lymphadenopathy.	Heart: moderate dilation of the RV and LV. Extensive organized thrombus in the CAs.	Heart: changes related to severe acute infarction. Necrotizing vasculitis of the CAs.	AMI
Tanimoto et al. 1981 [29]	1	8 y.o. M	In-life diagnosis. No treatment.	Sudden death 4 years after the onset of KD symptoms. The patient died while running.	Heart: LV wall showed white and dark-brown discoloration. Six CAAs at the LCA. After cutting the largest aneurysm, a red-brown fresh thrombus was found.	Heart: CAs arteriosclerotic changes with a thin elastic layer and intima thickening. CAs showed thrombosis. CAAs with fresh and organized thrombi. Old and acute LV infarction.	AMI
Font et al. 1983 [30]	1	4 m.o. M	In-life diagnosis. No treatment.	At the age of 3 months, the patient was hospitalized for otitis media, fever, and conjunctivitis. After discharge, he was hospitalized again because of pericardial effusion and some days later, after improvement, he became lethargic and suddenly died.	Heart: serosanguineous pericardial effusion. Vasculitis with thrombosis in the epicardial vessels. CAAs filled with thrombi. Other vessels: several, diffuse aneurysmal dilatations, some filled with thrombi. Other viscera: multiple infarcts of the spleen and kidneys, splenomegaly. Lymphoid hyperplasia of cervical axillary, and periportal lymph nodes. Subarachnoid hemorrhage, multifocal cerebellar hemorrhages, and microthrombi in several vessels of the thalamus. Necrosis of distal digits of the hands and feet.	Heart: vasculitis with thrombosis in the epicardial vessels. CAAs filled with thrombi. Other viscera: subarachnoid hemorrhage.	Cerebral hemorrhage
Imakita et al. 1984 [31]	3	3 m.o. F	In-life diagnosis. Treated with ASA.	She had a fever, bilateral conjunctival congestion, redness of the lips, and macular erythema. Three days later, she was hospitalized with a suspected diagnosis of KD. During her hospital stay, she was diagnosed with mitral insufficiency and irregularity of the coronary artery. Her persistent fever and general condition appeared to be improving during the hospital stay. The patient suddenly died 10 days after the onset of symptoms.	Heart: hypertrophy. Thickness of the LV wall. Multiple small, dome-like elevations on the mitral valve. The aortic valve was elongated and showed severe waving. Three CAAs (located at the proximal segments of the LCA and the RCA) filled with red-brown thrombi. Other viscera/vessels: fusiform aneurysms in the right internal iliac artery.	Heart: severe pancarditis. Edema and infiltration of lymphocytes and plasma cells around epicardial coronary arteries. Fibroblast proliferation and severe inflammatory infiltration in the endocardium. Edema and prominent infiltration of lymphocytes and large histiocytic cells in the myocardium, as well as small foci of degeneration and scarring. Valves: lesion composed of inflammatory infiltration with fibrous connective tissue, proliferation of small capillaries, and severe chronic inflammation of the cardiac valves. Perivascular inflammation and different stages of pathological changes in other visceral vessels.	Pancarditis
		4 m.o. M	In-life diagnosis. Treatment N/A.	Sudden death 21 days after the onset of KD.	Heart: hypertrophy. Serosanguinous hydropericardium. CA thrombosis. Other viscera: kidney infarction, microthrombosis in the lungs, liver and spleen congestion, brain edema. Other vessels: thrombosis in the iliac arteries. Other: hydrothorax, slightly turbid ascites.	Heart: myocarditis, pericarditis, valvulitis. CA thrombosis. Mild inflammatory infiltration, increment of fibrous connective tissue, and proliferation of small capillaries at the mitral valve. Mild inflammatory cell infiltration at the tricuspid valve.	Pancarditis
		2 m.o. M	In-life diagnosis. Treatment N/A.	Sudden death 33 days after the onset of KD.	Heart: hypertrophy. CA thrombosis. Other viscera: perivascular inflammation in the liver, kidneys, and lymph nodes. Infarction of the spleen and kidneys. Other vessels: diffuse artery thrombosis. Other: Ascites, dry gangrene of the fingers and toes.	Heart: pericarditis, endocarditis. CA thrombosis. Mild inflammatory infiltration and increment of fibrous connective tissue in the aortic and pulmonary valves. Other viscera: perivascular inflammation Other vessels: thrombosis.	Pericarditis
Embil et al. 1985 [32]	1	7 m.o. M	In-life diagnosis. Treatment N/A.	Four weeks after admission for aseptic meningitis (with fever, irritability, and maculopapular rash), the infant suffered from severe cardiac arrest.	Heart: multiple CAAs and thrombosis of the right coronary artery and the main branches of the LCA.	CA thrombosis.	AMI
Nakano et al. 1986 [33]	2	1 y.o. M	In-life diagnosis. Treatment N/A.	Sudden death 19 days after the onset of KD symptoms.	NA	NA	Arrhythmia
		1 y.o. M	In-life diagnosis. Treatment N/A.	Sudden death four months after the onset of KD symptoms. He died from severe hypoxic brain damage after cardiac arrest.	NA	NA	Hypoxic brain damage
Cloney et al. 1987 [34]	1	5 y.o. F	Postmortem diagnosis.	Hospitalized for fever and lymphadenitis. On the 21st day of hospitalization, she suddenly died.	Heart: all chambers were dilatated. Macroscopic evidence of myocardial infarction. CAAs (located at the LCA), filled with a nonoccluding thrombus.	Heart: recent (2–4-day-old) transmural myocardial infarction. Severe CA arteritis. Microemboli within the CA branches.	AMI
Fujiwara et al. 1988 [35]	5	10 m.o. M	In-life diagnosis. Treatment N/A.	Sudden death 9 days after the onset of KD.	Heart: Acute myocarditis.	Heart: acute myocarditis. Acute CA inflammation.	Myocarditis
		3 m.o. M	In-life diagnosis. Treatment N/A.	Sudden death 18 days after the onset of KD.	Heart: Acute myocarditis.	Heart: acute myocarditis. Acute CA inflammation. Slight dilatation of the major CAs.	Myocarditis
		1 y.o. M	In-life diagnosis. Treatment N/A.	Sudden death 19 days after the onset of KD.	Heart: Acute myocarditis.	Heart: acute myocarditis. Severe panvasculitis and severe inflammation of the perivascular area. Slight dilatation of the major CAs.	Myocarditis
		3 m.o. F	In-life diagnosis. Treatment N/A.	Sudden death 22 days after the onset of KD.	Heart: Acute myocarditis.	Heart: acute myocarditis. Slight microscopic mononuclear cell infiltration in the intima and adventitia of the CAs.	Myocarditis
		4 y.o. M	In-life diagnosis. Treatment N/A.	He died 2 days after the onset of a spasm of the LAD followed by acute myocardial infarction that occurred during cineangiography.	NA	Heart: Acute anteroseptal myocardial infarction. Abnormal fibrous intimal thickening of the CAs.	AMI
McCowen and Henderson 1988 [36]	1	12 y.o. M	Postmortem diagnosis.	Sudden death while walking.	Heart: CAAs of the LAD filled with thrombi. The distal LAD and the proximal portion of the RCA showed complete obliteration of the lumen by firm grayish tissue.	Heart: CAA wall showed a loss of smooth muscle and replacement by fibrous connective tissue. An organized and recanalized thrombus within the distal LAD. Other CAs showed extensive media fibrosis and elastic lamella disarray. Lungs: edema and congestion.	AMI
Sakai et al. 1988 [37]	1	39 y.o. M	In-life diagnosis. Treatment N/A.	Sudden death while carrying a refrigerator.	Heart: enlarged, LCA aneurysm, partly filled with a gelatin-like substance. Healed myocardial infarction. LAD occlusion.	Heart: the CAAs showed organized thrombi and calcification. Myocardial cells were highly fragmented and atrophied in the noninfarcted area. Healed myocardial infarction was confirmed.	ACF
Loubser et al. 1989 [38]	1	2 y.o. F	Postmortem diagnosis.	Hospitalized for palmar erythema, cervical lymphadenopathy, stomatitis, and mild hepatomegaly. Forty-eight hours later, the patient collapsed suddenly and could not be resuscitated.	Heart: RCA occluded by a recent thrombus; LAD wall thickened with a narrowed lumen.	Heart: Coronary arteritis.	AMI
Burke et al. 1990 [39]	1	17 y.o. M	Postmortem diagnosis.	Sudden death during exercise.	Heart: healed transmural posteroseptal infarction. Coronary arteries: aneurysm in the proximal segment of the RCA and in the LAD.	Heart: signs of myocardial infarction. CAA with mixed chronic inflammatory infiltrate in the wall.	AMI
Corradoet al. 1992 [40]	1	6 y.o. M	Postmortem diagnosis.	Sudden death 3 years after myocardial infarction.	Heart: CAAs and thrombosis, leading to severe ischemic cardiomyopathy.	NA	AMI
Naganuma et al. 1992 [41]	1	1 y.o. M	In-life diagnosis. Treated with ASA and IVIG.	Hospitalized for persistent fever, erythematous exanthema, convulsions, and vomiting. The patient recovered. Sudden death occurred 2 months after hospital discharge while the patient was asymptomatic.	Coronary arteries: in the RCA, two defective areas with fresh thrombi and marked stenosis in the RCA. Gradual narrowing of the LCA. Cervical lymph node and thymus swelling.	Heart: The CAs showed marked intima fibrocellular thickening, destruction of the internal elastic lamina and media, and marked periarterial fibrosis of the CAAs. Fresh thrombi within the CA lumen. The LV myocardium showed diffuse and transmural fresh myocardial necrosis. Other vessels: marked perivascular fibrosis with intimal thickening Immunohistochemical staining for desmin in the lesion of the left CAA showed many desmin-positive cells in the intima.	AMI
Shauka et al. 1993 [42]	1	24 y.o. M	Postmortem diagnosis.	He had chest pain, and myocardial infarction was diagnosed.	Heart: enlarged with LV hypertrophy. CAAs with an organized thrombus. Mural fibrosis.	NA	AMI
Smith and Grider 1993 [43]	1	18 y.o. M	Postmortem diagnosis.	He collapsed during a physical conditioning run.	Heart: thrombosed, fusiform CAAs.	Heart: CAAs with remote and recent thrombosis, calcification, and fibrotic changes. Remote transmural myocardial infarction of the LV and recent myocardial infarction of the RV.	Arrhythmia
Kristensen and Kristensen 1994 [44]	2	11 y.o. M	Postmortem diagnosis.	Found dead, no symptoms reported.	Heart: fibrous infarct in the septum. CAAs filled with thrombi.	Heart: CAAs showed intimal proliferative fibrosis, medial thinning with a loss of elastic membranes, and occlusive luminal thromboses. Extensive septal fibrosis.	Arrhythmia
		29 y.o. M	Postmortem diagnosis.	Sudden death.	Heart: enlarged with endocardial and myocardial fibrosis. Calcific CAAs (located at the LCA and RCA) with lumen occlusion.	Heart: CAAs with organized thrombus. Myocardium: extensive fibrosis	Arrhythmia
Lie and Sanders 1997 [45]	1	11 m.o. M	Postmortem diagnosis.	He presented at the ER with fever, irritability, difficulty breathing, and arching of the back. Cardiac arrest occurred within 1 h.	Heart: heavier than average. CAAs with luminal occlusion.	Heart: CAAs with intimal fibrocellular proliferation of organized thrombosis. Other CAs showed cord-like wall thickening. Recent multifocal and healed infarcts in the myocardium. Other vessels: iliac artery aneurysm-	AMI
Burke et al. 1998 [46]	2	4 y.o. M	Postmortem diagnosis.	Multiple hospital admissions for fever and abdominal pain. On the day of death, he complained of abdominal pain and became unresponsive.	Heart: diffuse CA thickening and stenosis.	Heart: fibrosis or necrosis of the LV lateral wall. Fibrointimal proliferation with focal destruction of the media and scattered inflammatory infiltrates in the intima, media, and adventitia of the CAs. Focal acute, organized thrombus.	AMI
		20 m.o. M	In-life diagnosis. Treated with ASA and IVIG.	At 8 m.o., he was diagnosed with KD. He had cyanosis and limpness, an ECG showed slight ST segment elevation or depression in most leads, and he died suddenly.	Heart: diffuse mottling. Diffuse intimal thickening and mild ectasia with pinpoint lumens of the CAs.	Heart: ongoing CAs arteritis with extensive fibrointimal proliferation and mixed inflammatory infiltrate. Occasional giant cells. Acute subendocardial infarction of the LV. Viscera: arteritis of the hepatic and renal arteries.	AMI
McConnell et al. 1998 [47]	1	3 y.o. M	In-life diagnosis. Treated with IVIG.	Death occurred 7 months after the onset of KD symptoms. Cardiac arrest during daily maintenance asthma treatment.	Heart: enlarged. Narrowing of the LCA, LAD, and RCA due to fibrosis.	Heart: fibrointimal proliferation with some smooth muscle proliferation of the CAs. Active inflammation with lymphocytes, plasma cells, and occasional eosinophils in all sections of the epicardial arteries.	ACS
Fineschi et al. 1999 [48]	1	21 y.o. M	Postmortem diagnosis.	Sudden death while playing soccer.	Heart: small white area of the internal part of the anterior wall of the LV. Calcified saccular CAAs (LAD and RCA) with pronounced stenosis distal to the aneurysms.	Heart: CAA walls showed an internal fibrocalcified layer and an external thin tunica media with an advanced vascularized and organized thrombus. The CA wall distal to the aneurysm showed obliterative intimal thickening (smooth muscle cell proliferation, elastic and fibrous network, and interstitial proteoglycan accumulation) with lumen narrowing. Old myocardial fibrosis with a few small islands of fatty tissue in the center. Foci of myocardial contraction band necrosis.	Arrhythmia
Suzuki et al. 1999 [49]	1	4 y.o. F	In-life diagnosis. Treatment NA.	Hospitalized for high fever with swollen lymph nodes, exanthema, strawberry tongue, and conjunctivitis. She died 18 days later, following the rupture of a large aneurysm of the LCA.	Heart: hemopericardium. Dilated LCA and RCA. Giant CAAs.	Heart: CAs vasculitis with proliferation of fibroblasts and myofibroblasts. The ruptured CAA showed panvasculitis accompanied by the destruction of elastic laminae.	CAA rupture
Kazuma et al. 2000 [50]	1	11 y.o. M	In-life diagnosis.	Sudden death while playing soccer.	Heart: thin at the apex of the LV. Ventricular wall fibrosis. Old myocardial infarction. Giant calcified CAA (LAD).	Heart: fibrosis, old myocardial infarction. Giant calcified aneurysm.	Arrhythmia
Maresi E. et al. 2001 [51]	1	2 m.o. M	Postmortem diagnosis.	Hospitalized for rhinitis, coughing, conjunctival hyperemia, and allergic exanthema. Sudden death after seven days.	Heart: LV hypertrophy. Hemopericardium. Perforated CAA of the LAD.	Heart: transmural lymphomonocytic inflammation of the CAA wall with some eosinophils, edema, and necrosis at the site of rupture. Chronic inflammation in the proximity of the epicardial coronary vessels. Small foci of active lymphocytic myocarditis.	CAA rupture
Heaton and Wilson 2002 [52]	2	9 m.o. M	In-life diagnosis. Treated with ASA and IVIG.	On day 95 after the beginning of KD symptoms, the patient died from myocardial infarction.	Heart: pale endocardium of the LV free wall and interventricular septum. Prominent segmental mural thickening and luminal stenosis of the CAs with mild CAAs of the LAD and RCA.	Heart: extensive patchy, acute subendocardial myocardial infarction. Circumferential mural fibrosis with chronic inflammatory cell infiltrate of the epicardial CAs, causing luminal stenosis.	AMI
		4 y.o. M	In-life diagnosis. Treated with ASA and IVIG.	LV function impairment after ten weeks of KD symptoms. He died after the induction of general anesthesia for cardiac catheterization.	Heart: thickening of the CA wall, LAD dilation.	Heart: fibrocellular intimal proliferation, causing severe luminal stenosis of all CAs. Myocardial infarction. Viscera: widespread systemic arteritis affecting other large- and middle-sized vessels with significant stenoses of the splenic, renal, superior, and distal mesenteric arteries. Duodenal infarction.	AMI
Bartoloni et al. 2002 [53]	1	21 y.o. M	Postmortem diagnosis.	Found dead on his bed after some days of fever, arthralgia, and emesis.	Heart: CAA of the CX with an occlusive recent thrombosis in the distal half of the artery.	Heart: foci of myocarditis. Posterior myocardial infarction. CA vasculitis and perivasculitis. The CAA wall showed polymorphous inflammatory infiltrate.	AMI
Rozin et al. 2003 [54]	1	21 y.o. M	Postmortem diagnosis	Found unresponsive on his bed.	Heart: LV hypertrophy. Focal myocarditis. Calcified CAA of the LAD, filled with a thrombus. Similar CAA in the CX. Minimal RCA aneurysmal dilation. Other organs: congestion and edema of the lungs, hepatosplenomegaly.	Heart: focal myocarditis. Large calcified fusiform CAAs filled with thrombi plus mural thickening.	ACS
Freema et al. 2005 [55]	8	10 y.o. M	In-life diagnosis. Treated with IVIG/ASA.	Sudden death 13 days after the onset of KD symptoms.	NA	Heart: microvessel density count showed grade 3 or 4 inflammation in the myocardium and CAA. The CAA wall was disrupted. VEGF, PDGF-A, and bFGF were detected diffusely in the myocardium. Angiostatin was detected predominantly in inflammatory cells in the CAA adventitia. Mast cells were present in the CAA adventitia and myocardium.	CAA rupture
		3 m.o. F	In-life diagnosis. No treatment.	Sudden death 14 days after the onset of KD symptoms.	NA		Myocarditis
		11 m.o. M	Postmortem diagnosis.	Sudden death 2 weeks after the onset of KD symptoms.	NA		CAA rupture
		4 m.o. M	In-life diagnosis. No treatment.	Sudden death 17–18 days after the onset of KD symptoms.	NA		AMI
		4 m.o. M	In-life diagnosis. Treated with IVIG/ASA.	Sudden death 3–4 weeks after the onset of KD symptoms.	NA		Myocarditis
		4 m.o. M	Postmortem diagnosis.	Sudden death 4 weeks after the onset of KD symptoms.	NA		ACF
		7 m.o. M	In-life diagnosis. No treatment.	Sudden death 4 weeks days after the onset of KD symptoms.	NA		AMI
		10 m.o. F	In-life diagnosis. No treatment.	Sudden death 5 weeks after the onset of KD symptoms.	NA		AMI
Tsuda et al. 2005 [56]	12	8 y.o. M	In-life diagnosis. No treatment.	Sudden death 4 years after KD onset.	Heart: coronary arterial lesions, acute and old myocardial infarction, CAAs, and severe localized stenosis of the LAD.	Heart: acute and old myocardial infarction. CA stenosis.	AMI
		1 y.o. M	In-life diagnosis. Treated with ASA and dipyridamole	Sudden death some months after KD onset and five days after cardiac catheterization.	Heart: acute inferior myocardial infarction with occlusion of the RCA.	Heart: acute and old myocardial infarction. CA stenosis.	AMI
		1 y.o. M	In-life diagnosis. Treated with ASA, nitrates, and CA antagonist	Sudden death more than 1 year after KD onset and 2 months after CA bypass grafting due to a distal aneurysm at the LAD.	Heart: patency of the grafts, but occlusion of the distal LAD.	CA stenosis.	AMI
		1 y.o. M	In-life diagnosis. Treated with flurbiprofen.	Sudden death some months after KD onset.	NA	NA	Undefined
		15 y.o. M	In-life diagnosis. Treated with ASA and dipyridamole.	Sudden death while playing soccer 4 years after KD onset. Diagnosed with segmental stenosis of the RCA and localized stenosis of the LAD.	NA	NA	Undefined
		18 y.o. M	In-life diagnosis. Treated with flurbiprofen.	Sudden death 12 years after KD onset.	NA	NA	AMI
		16 y.o. M	In-life diagnosis. Treated with ASA and warfarin.	Sudden death 5 years after KD onset. Diagnosed with a giant aneurysm of the LCA and occlusion of the RCA. He previously suffered from an inferior myocardial infarction.	NA	NA	Undefined
		22 y.o. M	In-life diagnosis. Treated with ASA.	Sudden death some months after KD onset. He previously suffered from myocardial infarction and was diagnosed with CX stenosis.	NA	NA	Undefined
		22 y.o. M	In-life diagnosis. Treated with nitrates and CA antagonist.	Sudden death six months after KD onset. Diagnosed with segmental stenosis of the LAD, CX, and RCA.	NA	NA	Undefined
		19 y.o. F	In-life diagnosis. Treated with flurbiprofen, a diuretic, and digoxin.	Sudden death six months after KD onset. She previously had a myocardial infarction.	NA	NA	Undefined
		27 y.o. F	In-life diagnosis. Treated with ASA, warfarin, ACE inhibitor, and beta-blocker D.	Sudden death about two years after the onset of KD symptoms. Diagnosed with myocardial infarction and mitral regurgitation and previously underwent coronary artery bypass grafting surgery and mitral annuloplasty.	NA	NA	Undefined
		26 y.o. M	In-life diagnosis. Treated with ASA and ACE inhibitor.	Found dead about 3 years after KD onset. He suffered from a previous myocardial infarction and LV dysfunction. He was diagnosed with segmental stenosis of the RCA.	NA	NA	Undefined
Ozdoguand Boga 2005 [57]	1	18 y.o. M	In-life diagnosis. Treated with IVIG.	He suffered from inflammatory myositis, lymphadenopathy, and mucosal changes. He died from massive hemorrhagic pericardial effusion and cardiac tamponade.	NA	NA	Cardiac tamponade
Diana et al. 2006 [58]	1	4 m.o. F	Postmortem diagnosis.	Previous episode of heart block. Sudden death shortly after recovery.	Heart: arteritis with a significant degree of aneurysmal dilatation of the LAD and RCA. Thrombosis.	CA arteritis. CAA with thrombosis.	ACS
Sunagawa et al. 2008 [59]	1	5 y.o. M	In-life diagnosis. Treatment N/A.	Sudden death following the rupture of a giant aneurysm of the LAD.	Heart: enlarged. Hemopericardium. CAA of the LAD with a fissure. Thrombus attached to the aneurysmal wall. CX and RCA dilation.	Heart: CAA and LAD CAA rupture, panvasculitis with inflammatory cell infiltration (monocytes, lymphocytes, plasmacytes, and macrophages), fibrinoid-like necrosis, and massive destruction of elastic fibers and smooth muscle cells. A small thrombus with focal organization in the aneurysmal wall. Inflammatory cells were seen in the immunohistochemical examination.	CAA rupture
Papadodima et al. 2009 [60]	1	11 y.o. M	Postmortem diagnosis.	Admitted to the hospital complaining from 5 days of fever and hematuria. After almost 2 weeks of hospitalization, he showed melena and intense abdominal pain and then suddenly died.	Heart: enlarged, intermediate edema. Multiple thrombi within the CAs.	Heart: intramural dense, polymorphonuclear inflammatory infiltration and necrosis of the CAs. CA thrombosis.	ACS
Yokouchi et al. 2010 [61]	1	40 y.o. F	Postmortem diagnosis.	Sudden death. He was a smoker and previously had an acute myocardial infarction that was treated with stent implantation. He developed multiple stent reocclusions.	Heart: Extensive fibrosis, fresh infarction in the anterior and posterior walls. CAAs in the RCA and LCA.	Heart: rupture of the internal elastic lamina of the LCA aneurysm, which had thrombi within the lumen. The RCA aneurysm was calcified with an organized thrombus. The lumen of each CA stent was occluded by thrombi and extensive inflammatory infiltration. Other CAs showed diffuse concentric intimal thickening, extension, and focal rupture of the internal elastic lamina, and focal destruction of the tunica media. Aorta: slight fibrotic thickening of the intima, few atherosclerotic changes.	Arrhythmia
Pucci et al. 2012 [62]	2	3 m.o. F	Postmortem diagnosis.	Sudden cardiac arrest after eight days of intermittant fever, red and cracked lips, maculopapular rash.	Heart: pericardial effusion. CAAs with occlusive thrombosis of the LAD.	Heart: CA thrombosis associated with inflammatory infiltration in the CAA wall. Contraction bands and multiple foci of T-lymphocytic myocarditis.	AMI
		3 m.o. M	Postmortem diagnosis.	Sudden cardiac arrest after ten days of intermittant fever, bronchiolitis, and exanthema.	Heart: moderate pericardial effusion. Ectatic cord-like shaped coronary arteries with subocclusive or laminar thrombosis and multiple CAAs.	Heart: CAA thrombosis associated with inflammatory infiltrate. Multiple foci of coagulative necrosis and diffuse T-lymphocytic infiltrates.	AMI
Ponniah 2013 [63]	1	6 m.o. M	Postmortem diagnosis.	Diagnosed with CAA due to KD, died from extensive CA thrombosis.	Heart: multifocal myocardial scars. CAAs with occluding thrombus.	Heart: CAAs with thrombi and intimal wall thickening.	ACS
Okura et al. 2013 [64]	1	30 y.o. M	In-life diagnosis. Treatment N/A.	Sudden death. Previously diagnosed with acute KD and a complicated CAA.	Postmortem CT: coarse calcification of the CA aneurysm	Thin CAA wall with myointimal proliferation associated with a disrupted internal elastic lamina and medial smooth muscle necrosis with replacement by fibrocalcification.	ACS
Miyamoto et al. 2014 [65]	1	3 m.o. M	In-life diagnosis. Treated with IVIG and prednisolone sodium metazoate.	Sudden death due to CAA rupture.	RCA aneurysm rupture combined with multiple CAAs in the CX and LAD.	CAA: rupture. Thickened aneurysmal wall with the presence of inflammation.	CAA rupture
Shimiz et al. 2015 [66]	2	22 y.o. M	In-life diagnosis. Treated with IVIG twice, followed by ASA and dipyridamole	Sudden death	Heart: enlarged, smooth epicardium. Pale myocardial areas. CAA of the LAD. The distal artery was very small and partially occluded.	Heart: Adventitial sparse cellular fibrosis, extensive dystrophic calcification, and regions of chondro-ossification of the LAD. Patchy myocardium, widespread fibrosis with compensatory hypertrophic changes, acute ischemia areas. Contraction band necrosis. Other viscera: congestion with marked pulmonary edema and congestion.	AMI
		30 y.o. M	Postmortem diagnosis.	Found unresponsive.	Heart: enlarged, smooth epicardium, posterior apical myocardium infarct with patchy interstitial fibrosis and pink discoloration. CAA of the LAD, which had a thin and calcified wall and a thrombus within the lumen. Re-canalized CAA with multiple lumina of the RCA.	Heart: focal interstitial fibrosis and scattered thickened myocytes with enlarged nuclei. Asymmetric transmural fibrosis of the LAD with dystrophic calcification, causing eccentric narrowing of the lumen that was occluded by an acute thrombus. The CAA wall showed a small focus of calcification, diffuse fibrosis, and a disrupted internal elastic lamina.	AMI
Parsons and Lynch 2016 [67]	1	22 y.o. M	In-life diagnosis. Treatment N/A.	Sudden death.	Heart: enlarged, LV hypertrophy, myocardial fibrosis, probe patent foramen ovale. CAA of the LAD.	Heart: myocardial fibrosis. CA stenosis.	ACS
Chang et al. 2016 [68]	1	6 y.o. M	Postmortem diagnosis.	Suddenly died while experiencing nausea, feeling unwell, and dizziness.	Heart: CAAs. A recent thrombus within a CAA lumen.	CAA: focal but minimal lymphoplasmacytic infiltration in the LCA and myofibroblastic proliferation within the LAD, CX, and RCA. Heart: acute myocardial infarction, partly transmural, affecting the anteroseptal and anterolateral LV. An old healed subendocardial infarct along the basal half of the lateral wall of the LV and patchy subendocardial fibrosis affecting the posteroseptal wall of the LV.	AMI
Wei et al. 2016 [69]	7	7 y.o. M	In-life diagnosis. Treated.	Chest pain, abdominal pain, vomiting, and dyspnea and then died four years after the onset of KD symptoms.	4 CAs involved.	NA	AMI
		9 y.o. M	In-life diagnosis. Treated.	Chest pain, abdominal pain, vomiting, and dyspnea and then died two years after the onset of KD symptoms.	3 CAs involved.	NA	AMI
		8 y.o. F	In-life diagnosis. Treated.	Chest pain, abdominal pain, vomiting, and dyspnea and then died four years after the onset of KD symptoms.	2 Cas involved.	NA	AMI
		4 y.o. M	In-life diagnosis. Treated.	Sudden death 3 years after the onset of KD symptoms.	2 CAs involved.	NA	Undefined
		17 y.o. F	In-life diagnosis. Treated.	Sudden death 13 years after the onset of KD symptoms.	3 CAs involved.	NA	Undefined
		10 m.o. M	In-life diagnosis. Treated.	Sudden death 5 months after the onset of KD symptoms.	1 CA involved.	NA	CAA rupture
		1 y.o. M	In-life diagnosis. Treated.	Sudden death 29 days after the onset of KD symptoms.	Heart: pancarditis and hemopericardium. CAAs with thrombosis. Other organs: multiple small arteries and veins were involved.	Heart: pancarditis. CAA rupture. CA thrombosis.	CAA rupture
Yajima et al. 2016 [70]	1	5 m.o. M	Postmortem diagnosis.	Hospitalized for fever and suppuration at the site of (BCG) vaccination. He suddenly died after hospital discharge.	Heart: fluid in the pericardium. Epicardium petechiae. Other viscera: pleural fluid, fluid in thetracheal space, pleural petechiae. Small subcapsular liver hemorrhages. Brain edema.	Heart: myocardial and CA inflammatory infiltrations (mononuclear cells, mainly lymphocytes). Emboli within the CAs. Other viscera: inflammation of the lungs, liver, and kidneys.	AMI
Fukazawa et al. 2017 [71]	11	N/A	5 cases, in-life diagnosis. Treated with IVIG.	Death due to CA rupture within 1 month from the onset of KD symptoms.	NA	NA	5 CAA rupture
		N/A	6 cases, in-life diagnosis. Treatment N/A.	Most deaths due to MI occurred from 6 months to 2 years from the onset of KD symptoms.,	NA	NA	6 AMI
Kim et al. 2018 [72]	1	23 m.o. M	Postmortem diagnosis.	Sudden death 6 weeks after an episode of mild fever, general myalgia, and diarrhea.	Heart: pericardial yellowish effusion. Myocardium infarction, both recent and old. CAAs.	Heart: foci of acute and old infarction. The LAD wall showed mural lymphoplasmacytic infiltration, adventitial fibrosis, and neovascularization. Polymorphonuclear leukocyte and lymphocyte infiltration with focal acute necrosis, which resulted in the destruction of the internal elastic lamina at the RCA.	AMI
Zhang et al. 2018 [73]	1	5 y.o. M	Postmortem diagnosis	Sudden death due to the rupture of a CAA during the acute phase of KD.	Heart: hemopericardium. Ruptured CAA of the LAD.	Heart: granulation tissue in cardiac adventitia. Necrotic cardiac muscle fibers in the myocardium. The CAA showed transmural inflammation, disruptions of the internal elastic lamina, smooth muscle, and inflammatory infiltrates with edema and variable necrosis of the intima, media, and adventitia. Mixed thrombosis in the aneurysmal lumen.	CAA rupture
Pachec et al. 2019 [74]	1	13 y.o. M	Postmortem diagnosis	Sudden death 2 weeks after streptococcal pharyngitis.	Heart: enlarged, mild epicarditis, pale myocardium. CA luminal stenosis.	Heart: globally edematous myocardium. CA wall showed inflammatory infiltrates, muscular-layer destruction, and a loss of elastic tissue. Some vessels had recent thrombi.	ACS
Flossdorf et al. 2020 [75]	1	20 y.o. M	Postmortem diagnosis.	Found dead after vomiting the day before.	Heart: ventricular aneurysm below the aortic valve. Multiple myocardial scars. CAAs filled with thrombi.	Heart: the CAA wall showed thickening of the intima, thin media with only a small residual amount of smooth muscle cells, and transmural inflammation with circumscribed calcified plaques at the border to the innermost layer of the arteries. Neoangiogenesis around the vessels. CA thrombosis and stenosis.	ACS
Zhang and Wang 2021 [76]	1	1 y.o. M	Postmortem diagnosis	Hospitalized for fever and cough with a preliminary diagnosis of acute severe bronchial pneumonia but no typical KD characteristics. After antibiotics and supportive treatment, the condition worsened and then he died.	Heart: cable-shaped bulge at the anterior area of the heart.	Coronary artery: inflammatory granulation tissue. Wall thrombosis and LCA malformation accompanied by vasculitis.	ACS
Staats et al. 2021 [77]	1	4 y.o. F	In-life diagnosis. Treated with warfarin, atorvastatin, and ASA.	Sudden death. Previously diagnosed with CAAs.	Heart: diffuse patchy, pale areas of scar tissue in the interventricular septum, associated with severe ischemic changes of cardiomyocytes. Luminal obliteration of her coronary aneurysms with organized thrombi.	Heart: myocardial fibrosis, ischemic changes. CA stenosis and thrombosis.	Arrhythmia
Maeda et al. 2021 [78]	1	2 y.o. F	In-life diagnosis. Treated with ASA and IVIG.	Hospitalized for fever and cervical lymphadenopathy. On day 12 after the beginning of symptoms, she vomited and had generalized tonic–clonic seizures without recovery of consciousness. Cranial MRI showed vasogenic edema. She died in the following hours.	NA	NA	Acute encephalitis
Ayusawa et al. 2022 [79]	14	School-aged F	In-life diagnosis.	Sudden death. Previously diagnosed with KD and CABG.	NA	NA	Undefined
		School-aged F	In-life diagnosis.	Sudden death. Previously diagnosed with KD and AMI.	Heart: fibrosis at the posterior and lateral walls of the LV. Dilative hypertrophy of the RV, Weight of the heart: 340 g. Coronary arteries: LCA aneurysm with total occlusion, RCA aneurysm.	Heart: myocardial fibrosis.	AMI
		School-aged M	In-life diagnosis.	Sudden death. Previously diagnosed with KD and CAA.	NA	NA	Undefined
		School-aged F	In-life diagnosis.	Sudden death. Previously diagnosed with KD and CAAs.	NA	NA	Undefined
		School-aged M	In-life diagnosis.	Sudden death. Previously diagnosed with KD and CAA.	NA	NA	AMI
		School-aged M	In-life diagnosis.	Sudden death. Previously diagnosed with KD.	NA	NA	ACF
		School-aged M	In-life diagnosis.	Sudden death. Previously diagnosed with KD.	NA	NA	AMI
		School-aged M	In-life diagnosis.	Sudden death. Previously diagnosed with KD.	NA	NA	AMI
		School-aged M	In-life diagnosis.	Sudden death. Previously diagnosed with KD and CAA.	Heart: ischemic heart disease. CA atherosclerosis, RCA aneurysm.	NA	Undefined.
		School-aged M	In-life diagnosis.	Sudden death. Previously diagnosed with KD.	NA	NA	ACF
		School-aged M	In-life diagnosis.	Sudden death. Previously diagnosed with KD.	NA	NA	AMI
		School-aged M	In-life diagnosis.	Sudden death. Previously diagnosed with KD.	NA	NA	Arrhythmia
		School-aged M	In-life diagnosis.	Sudden death. Previously diagnosed with KD and CAA.	NA	NA	AMI
		School-aged M	In-life diagnosis.	Sudden death. Previously diagnosed with KD.	NA	NA	Arrhythmia

ACF = Acute Cardiac Failure; ACS = Acute Coronary Syndrome; AMI = Acute Myocardial Infarction; CA = Coronary Artery; CAA = Coronary Artery Aneurysm; CABG = Coronary Artery Bypass Graft; CX = Circumflex Coronary Artery; ER = Emergency Room; F = female; ICU = Intensive Care Unit; IVIG = Intravenous Immunoglobulin; KD = Kawasaki Disease; LAD = Left Anterior Descending Coronary Artery; LCA = Left Coronary Artery; LV = Left Ventricle; m.o. = months old; y.o. = years old; M = male; NA = Not Available or Not Applicable; RCA = Right Coronary Artery; w.o. = weeks old.

**Table 2 diagnostics-13-01831-t002:** This table shows the main characteristics of the articles included in this review. NA = not available.

Characteristics			Number of Cases (tot = 117)
Sex	Male		86 (73.50%)
	Female		20 (17.09%)
	NA		11 (9.40%)
Age (y.o.)	<5		28 (23.93%)
	6–12		26 (22.22%)
	13–20		27 (23.08%)
	>20		10 (8.55%)
	NA		15 (12.82%)
Diagnosis	In-life (*n* = 80)	Treated	37 (31.63%)
		Not treated	7 (5.98%)
		Unknown	36 (30.77%)
	Postmortem		37 (31.62%)

**Table 3 diagnostics-13-01831-t003:** This table shows the causes of death for the reviewed cases.

Characteristics		Cause of Death												
		AMI	ACF	Peri-carditis	Pan-carditis	Myo-Carditis	Arrhythmia	ACS	CAA Rupture	Cardiac Tamponade	Acute Encephalitis	Cerebral Hemorrhage	Cerebral Hypoxia	Undefined
Sex	Male	36	6	2	1	4	8	9	8	1	-	1	1	9
	Female	6	1	-	1	2	2	1	1	-	1	-	-	5
	Unknown	6	-	-	-	-	-	-	5	-	-	-	-	-
Age (y.o.)	<5	9	3	2	2	5	-	2	4	-	-	1	-	-
	6–12	12	1	-	-	1	2	2	4	-	1	-	1	2
	13–20	15	2	-	-	-	4	1	1	-	-	-	-	4
	>20	2	-	-	-	-	1	2	-	1	-	-	-	4
	N/A	4	1	-	-	-	3	3	-	-	-	-	-	4
Total		48 (41.03%)	7 (5.98%)	2 (1.71%)	2 (1.71%)	6 (5.13%)	10 (8.55%)	10 (8.55%)	14 (11.97%)	1 (0.85%)	1 (0.85%)	1 (0.85%)	1 (0.85%)	14 (11.97%)

**Table 4 diagnostics-13-01831-t004:** This table shows the main gross findings for the reviewed cases. Only cases in which autopsy data were reported were taken into consideration.

Autoptic Data Gross Findings			
	Cardiac Death	Cerebral Death	Undefined Death
Enlarged heart	19	-	-
Heart hypertrophy	4	-	-
Hemopericardium	3	-	-
Pericardial effusion	7	1	-
Myocardial fibrosis	21	-	-
Coronary thrombosis *	7	1	-
Coronary stenosis *	18	-	-
CAA **	17	-	1
CAAs ***	29	1	-
CAA filled with thrombus	22	1	-
CAA rupture	14	-	-
Aneurysms of other vessels	1	-	-
Thrombosis of other vessels	2	1	-
Other visceral involvement	1	1	-
Other visceral congestion	4	-	-
Subarachnoid hemorrhage	-	1	-
Total cases	65	1	1

* In the nonaneurysmatic region, if present. ** One artery involved. *** More than one artery involved.

**Table 5 diagnostics-13-01831-t005:** This table shows the main histological findings for the reviewed cases. Only cases in which microscopic data were reported were taken into consideration.

Histopatological Findings		
	Cardiac Death	Cerebral Death
Acute myocardial infarction	17	-
Myocardial fibrosis	17	-
Myocardial necrosis	11	-
Myocardial inflammatory infiltrates	14	-
Pericardial inflammatory infiltrates	1	-
Peri-, myo-, and endocardium inflammatory infiltrates	3	-
Coronary artery stenosis	16	-
Coronary artery fibrosis	13	-
Coronary artery thrombosis	19	1
Coronary thrombus recanalization	1	-
Coronary artery wall inflammatory infiltrates	26	-
Coronary artery wall thickening	9	-
Coronary artery wall necrosis	4	-
Coronary wall thinning	3	-
CAA thrombosis	17	1
CAA wall fibrosis	10	-
CAA wall calcification	7	-
CAA thrombus recanalization	2	-
CAA rupture	7	-
Neoangiogenesis	4	-
Systemic vessels wall inflammatory infiltrates	7	-
Systemic vessels thrombosis	2	1
Kidney infarction	1	-
Renal artery stenosis	1	-
Cerebral hemorrhage	-	1
Total cases	71	1

## Data Availability

No new data were created or analyzed in this study. Data sharing is not applicable to this article.

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
