# Peer review of "Autoptic Findings in Cases of Sudden Death Due to Kawasaki Disease"

_diagnostics, 2023, doi:10.3390/diagnostics13111831_

Round 1

Reviewer 1 Report

The article is quite interesting. I would like however to make the following comments/suggestions:

Lines 22-23: “Our work revealed that only a few cases, if compared with the incidence of KD, suffered from sudden death, and underwent the autoptic examination.”

Lines 158-159: “Unfortunately, only 67 and 72 cases have been completely described in the Literature, with autoptic and histopathological reports respectively

Lines 164-165: “Indeed, only a few autoptic reports are present in the recent Literature, as the pathology is, to date, rarely fatal.”

Lines 170-171: “The present systematic review revealed that only a few cases, if compared with the number of people affected by KD, underwent the autoptic examination and then were described in the Literature, mainly through case reports or series.”

In the literature cases or case series with new findings are usually reported. Consequently, not all deaths due to Kawasaki disease submitted to autopsy have been described in the literature. The above sentences should be modified accordingly in order to avoid misinterpretation. Moreover, 72 cases described in the literature is not a small number.

Lines 54-55: “when any other explanation for the illness 54 is possible”/is not possible

Line 94: “cases of survival patients” /survived patients

I

Given that the authors have reviewed the cases of death due to Kawasaki disease reported in the literature, I believe that some discussion on mortality data from previous studies would be useful. Examples of studies:

Nakamura Y, Yanagawa H, Harada K, Kato H, Kawasaki T. Mortality among persons with a history of Kawasaki disease in Japan: the fifth look. Arch Pediatr Adolesc Med. 2002 Feb;156(2):162-5. doi: 10.1001/archpedi.156.2.162. PMID: 11814378.

Makino N, Nakamura Y, Yashiro M, Sano T, Ae R, Kosami K, Kojo T, Aoyama Y, Kotani K, Yanagawa H. Epidemiological observations of Kawasaki disease in Japan, 2013-2014. Pediatr Int. 2018 Jun;60(6):581-587. doi: 10.1111/ped.13544. PMID: 29498791.

Manlhiot C, O'Shea S, Bernknopf B, LaBelle M, Chahal N, Dillenburg RF, Lai LS, Bock D, Lew B, Masood S, Mathew M, McCrindle BW. Epidemiology of Kawasaki Disease in Canada 2004 to 2014: Comparison of Surveillance Using Administrative Data vs Periodic Medical Record Review. Can J Cardiol. 2018 Mar;34(3):303-309. doi: 10.1016/j.cjca.2017.12.009. Epub 2017 Dec 15. PMID: 29395706.

Although I am not the most qualified person to judge, I believe that the quality of the English language in the manuscript is quite good. 

Lines 54-55: “when any other explanation for the illness 54 is possible”/is not possible

Line 94: “cases of survival patients” /survived patients

In the table “Papadomima”/Papadodima

Line 128: “37 cases had a post-mortem diagnosis”/The phrase should not begin with a number. A total number of 37 cases….. or Thirty-seven cases…..

Line 126: “80 subjects had an in-life diagnosis”/ the same

Why “literature” and “authors” are written in the text with L and A capitals?

 Lines 182-183: “Two decades after its discovery, KD surpassed rheumatic fever as the most common causes of acquired heart disease in children in Asia”/cause

 Line 271: “in such patients”/in those patients

Author Response

Dear reviewer, thank you for your comments. We made the changes you suggested:

“Lines 22-23: “Our work revealed that only a few cases, if compared with the incidence of KD, suffered from sudden death, and underwent the autoptic examination.”

Lines 158-159: “Unfortunately, only 67 and 72 cases have been completely described in the Literature, with autoptic and histopathological reports respectively”

Lines 164-165: “Indeed, only a few autoptic reports are present in the recent Literature, as the pathology is, to date, rarely fatal.”

Lines 170-171: “The present systematic review revealed that only a few cases, if compared with the number of people affected by KD, underwent the autoptic examination and then were described in the Literature, mainly through case reports or series.”

In the literature cases or case series with new findings are usually reported. Consequently, not all deaths due to Kawasaki disease submitted to autopsy have been described in the literature. The above sentences should be modified accordingly in order to avoid misinterpretation. Moreover, 72 cases described in the literature is not a small number”.

We made the changes you asked.

“Lines 54-55: “when any other explanation for the illness 54 is possible”/is not possible

Line 94: “cases of survival patients” /survived patients”

 We made the changes.

“Given that the authors have reviewed the cases of death due to Kawasaki disease reported in the literature, I believe that some discussion on mortality data from previous studies would be useful. Examples of studies:

Nakamura Y, Yanagawa H, Harada K, Kato H, Kawasaki T. Mortality among persons with a history of Kawasaki disease in Japan: the fifth look. Arch Pediatr Adolesc Med. 2002 Feb;156(2):162-5. doi: 10.1001/archpedi.156.2.162. PMID: 11814378.

Makino N, Nakamura Y, Yashiro M, Sano T, Ae R, Kosami K, Kojo T, Aoyama Y, Kotani K, Yanagawa H. Epidemiological observations of Kawasaki disease in Japan, 2013-2014. Pediatr Int. 2018 Jun;60(6):581-587. doi: 10.1111/ped.13544. PMID: 29498791.

Manlhiot C, O'Shea S, Bernknopf B, LaBelle M, Chahal N, Dillenburg RF, Lai LS, Bock D, Lew B, Masood S, Mathew M, McCrindle BW. Epidemiology of Kawasaki Disease in Canada 2004 to 2014: Comparison of Surveillance Using Administrative Data vs Periodic Medical Record Review. Can J Cardiol. 2018 Mar;34(3):303-309. doi: 10.1016/j.cjca.2017.12.009. Epub 2017 Dec 15. PMID: 29395706.

We added the studies you suggested and implemented the section on mortality rate after the acute phase of KD.

“Comments on the Quality of English Language

Although I am not the most qualified person to judge, I believe that the quality of the English language in the manuscript is quite good. 

Lines 54-55: “when any other explanation for the illness 54 is possible”/is not possible

Line 94: “cases of survival patients” /survived patients

In the table “Papadomima”/Papadodima

Line 128: “37 cases had a post-mortem diagnosis”/The phrase should not begin with a number. A total number of 37 cases….. or Thirty-seven cases…..

Line 126: “80 subjects had an in-life diagnosis”/ the same”

Thank you for the suggestions, we made all the changes.

“Why “literature” and “authors” are written in the text with L and A capitals?”

Sorry for the typing error, we corrected them.

“Lines 182-183: “Two decades after its discovery, KD surpassed rheumatic fever as the most common causes of acquired heart disease in children in Asia”/cause

 Line 271: “in such patients”/in those patients”

We made the changes.

Reviewer 2 Report

The authors present an interesting review article. The English is good, although they could consider a check from a native speaker.

In Mat. and Meth. the authors state that ‘papers presenting cases where death occurred after a long hospital stay’ were excluded. They should define the time period of ‘long hospital stay ’and indicate the arguments why studies with a long history of KD without hospitalization were included, whereas those with a ‘long hospital stay’ were excluded. In other words these exclusion and inclusion criteria should be explained in more detail.

A minor comment regarding the discussion – autopsy can also provide information regarding the effect of the applied, potentially novel therapy. The authors might consider this an another argument why autopsy should be performed even in cases with a well-documented clinical history.

check by a native speaker is recommended

Author Response

Dear reviewer, thank you for your comments. We made the exclusion criteria clearer as requested. Thank you for the suggestion on a possible new point of view for the postmortem research on KD patients. We implemented the end of the discussion.

Reviewer 3 Report

 The authors collected the literature on sudden deaths from Kawasaki disease and reviewed autoptic findings. The study of 117 cases reported in 54 papers is an impressive effort. However, the lack of novelty is a problem, making it difficult to evaluate as a new article.

 When rewriting the paper, a new point of view that has not been presented in the past is needed. It would be good to focus on differences by country or race; compare Asian and Western countries, or to compare Asian, Caucasian, and Negro. If they examine changes over time, it is possible that cardiac deaths have decreased in recent years due to advances in treatments. Detailed analysis of cases other than cardiac deaths is interesting, but this may be difficult because of the small number of cases.

None

Author Response

Dear reviewer, thank you for your comments. The purpose of the review is to search the literature for sudden or unexpected deaths due to KD. As you acknowledged, 117 cases are not a few, but we believe they are still not enough to be able to carry out relevant statistical analyses. We paid attention to the aspects you suggested and made some changes. Unfortunately, we do not have much data to focus on the comparison among Asian, Caucasian and Negro patients. However, the other two aspects you underlined (comparison on incidence among Asian and Western countries and response to therapy over time and decrease of sudden death), are mentioned in lines 184-189 and 190-193. Thanks to your comments we have further emphasized their importance.

Furthermore, we have highlighted, as a possible new point of view, the need to carry out the autoptic examination also for subjects who have an in-life diagnosis of KD, in order to implement the knowledge of the pathology, especially from a histological point of view, for a better understanding of the molecular pathogenetic mechanisms. Especially in the light of new scientific research technologies. Moreover, we suggested to conduct postmortem investigations to improve understanding of the effectiveness of the various treatment options.

Round 2

Reviewer 3 Report

There were no novel findings.

Author Response

Dear reviewer, 

we implemented the discussion section, adding also a sentence about the limits of the study, as the academic editor suggested.